# Multi-Site and Multi-Year Remote Records of Operative Temperatures with Biomimetic Loggers Reveal Spatio-Temporal Variability in Mountain Lizard Activity and Persistence Proxy Estimates

**Florèn Hugon [1,*]**, **Benoit Liquet [1,2]** and **Frank D'Amico [1]**

1 Laboratoire de Mathématiques et de Leurs Applications de Pau, UMR 5142, UFR des Sciences et Techniques de la Côte Basque, Allée du Parc Montaury, CNRS/Université Pau et Pays Adour/E2S UPPA, 64600 Anglet, France; benoit.liquet@univ-pau.fr (B.L.); frank.damico@univ-pau.fr (F.D.)
2 Department of Mathematics and Statistics, Macquarie University, Sydney, NSW 2109, Australia
\* Correspondence: floren.hugon@univ-pau.fr

**Abstract:** Commonly, when studies deal with the effects of climate change on biodiversity, mean value is used more than other parameters. However, climate change also leads to greater temperature variability, and many papers have demonstrated its importance in the implementation of biodiversity response strategies. We studied the spatio-temporal variability of activity time and persistence index, calculated from operative temperatures measured at three sites over three years, for a mountain endemic species. Temperatures were recorded with biomimetic loggers, an original remote sensing technology, which has the same advantages as these tools but is suitable for recording biological organisms data. Among the 42 tests conducted, 71% were significant for spatial variability and 28% for temporal variability. The differences in daily activity times and in persistence indices demonstrated the effects of the micro-habitat, habitat, slope, altitude, hydrography, and year. These observations have highlighted the great variability existence in the environmental temperatures experienced by lizard populations. Thus, our study underlines the importance to implement multi-year and multi-site studies to quantify the variability and produce more representative results. These studies can be facilitated by the use of biomimetic loggers, for which a user guide is provided in the last part of this paper.

**Keywords:** activity time; biomimetic model; biomimetic logger; data logger; operative temperature; persistence; remote sensors; spatio-temporal variability

## 1. Introduction

### 1.1. Climate Change and Spatio-Temporal Variability

Biodiversity conservation is a crucial issue related to climate change. To improve management measures, it is necessary to broaden our ecosystem knowledge [1]. Commonly, when studies deal with the statistical effects of climate change on biodiversity, the common and oversimplified way of addressing them is to rely on the mean despite climate change also leads to greater temperature variability and is associated with higher average climate parameters with greater variability [2]. The amplitude of daily temperatures is often greater and extreme events may be more frequent [3–5]. For instance, in France, the end of February 2019 was one of the hottest in history [6]. In 2020, the same exceptional heat was observed at the beginning of the month and there was also a snow cover deficit [7]. However, typically, the literature on the impact of climate change on biodiversity addresses the issue

often in terms of the mean and ignores variability in time and/or in space [8–10]. Many studies acknowledge spatial variability and use spatial replication but eventually average the results from the different study sites (see in [11] for discussion). Very few address temporal variability; experiments are very often conducted over only one year (see, for example, in [12]). Absence of true temporal and/or spatial replication to address explicitly variability issues is often linked to budget limitations but it still raises the crucial issue of the representativeness of the results. Many other studies demonstrate indeed the important role of variability in explaining species responses to climate change [13–15] as responses appear to be favored by thermal regimes more than by the mean temperature itself [16,17]. Moreover, several studies have shown that there are interactions between mean and variability. For example, these two parameters better explain thermal performance when considered together in the model rather than separately as they act in synergy [10]. Temperature variability has different impacts depending on the mean temperature and the thermal performance curve. If the mean is low, the temperature variability increase may lead to temperatures allowing a better performance; otherwise, if the average is already close to the optimum temperature, it will lead to a decrease in performance [8,18]. Furthermore, increased variability may cause heat stress leading to higher energy costs for the maintenance of individuals, which may consequently reduce reproductive performance [9,19]. Climate variability appears to play a major role in species persistence, so it is crucial to integrate it into studies dealing with climate change effects on biodiversity [20,21].

### 1.2. Spatio-Temporal Studies Using Biomimetic Logger, an Original Remote Sensing Tool

Quantifying variability requires that data are collected at multiple spatial and temporal scales. However, designing spatio-temporal research is often challenging for practical (field constraints) and technical reasons because analyzing spatially explicit and temporally resolved datasets can be tricky; the data are by essence autocorrelated and thus violate the basic assumption of independence. To reduce constraints and facilitate the design and practical implementation of any field study in the specific context of mountain landscape, we can first start by reducing the time spent in the field, thus lowering human and material resources costs, by using more appropriate tools [22].

In the broadest sense, remote sensing tools allow the remote acquisition of data concerning an object without contact with it [23–25]. These tools have many advantages, particularly interesting for the spatio-temporal studies implementation [26]. Remote acquisition allows to reduce the field constraints related to the study sites isolation, their difficult access, the climatic conditions, *etcetera* [23,27]. Moreover, they are well adapted to provide high-frequency and high-quality datasets with greater time-consumption efficiency [22,28]. Moreover, they measure data always in the same way in time and space, which allows for high reproducibility of studies [26,29–31]. Conventional remote sensing tools such as drones or satellites acquire data from observing the Earth's surfaces by recording the electromagnetic radiation emitted or reflected by the landscape [24]. Collected data are mainly used to study plant biodiversity—land use, land cover, deforestation [32–34], and climate—land surface temperature, and climatic risks [35,36]. Animal biodiversity, which is comparatively still little studied with remote sensing tools [34], can also benefit from the advantages of these tools. In particular, the biomimetic loggers use allows the acquisition of data specific to biological organisms [11].

A biomimetic logger [37–39], also called a biomimetic sensor [40,41], is composed of a data-logger and a biomimetic model. A data-logger is an electronic device coupled with at least one probe, also called a sensor, that automatically measures physical quantities at regular time intervals [23,42]. It is a discrete, easy-to-use, miniaturized device that produces reliable high-frequency data [43,44]. The small size of the probes allows their insertion into biomimetic models that mimic the organism physical properties [45–47]. Biomimetic logger allows the measurement of different reliable proxies [40], a proxy being a measurable variable that correlates with an unmeasurable variable [48]. For example, the biomimetic logger use composed of a data-logger equipped with a thermal probe inserted in a model mimicking the thermal properties of an organism permits the recording of a proxy of its body temperature [37]. This set is deployed in the field, the temperatures are recorded independently

at regular time intervals and stored in the data-logger until they are collected. In the sense that remote sensing refers to the remote acquisition of data concerning an object via noncontact with it [24,25], biomimetic loggers can be considered as remote sensing tools. Like other remote sensing technologies, it also enables low-cost monitoring while ensuring high quality of recorded data [49,50]. Unlike traditional tools, data acquisition requires the equipment installation and then, most often, a return to the field to collect the stored data. At the moment, no other tool allows efficient remote collection of biological data without contact with the studied organism. Thus, the biomimetic logger use is interesting to provide essential data for biodiversity conservation studies [51].

Data acquisition with this biomimetic logger is a highly valued noninvasive technique for the study of threatened and secretive species. Their use rather than deploying data loggers directly on the animals permits the organism study without capturing them, thus avoiding a major stress source and the capture effect on the measurements, which would have to be a posteriori corrected [40,52]. In addition, continuous independent recording allows studies to be conducted in harsh environments where study sites are difficult to access in terms of localization and/or legislation [53,54]. This tool also facilitates long-term data acquisition by drastically reducing the duration of field campaigns [37,55]. Moreover, the recorded data are more realistic than laboratory measurements, more accurate, and organism-specific [56,57]. For instance, the authors of [38] demonstrated a difference of 11.1 °C between micro-habitat temperature—used as a proxy of body temperature—measured with a weather station and body temperature measured with a biomimetic logger. This technology captures the proximal environment effect, such as soil moisture and wind speed, which may also influence body temperature [58]. Biomimetic logger monitoring is frequently used in thermal physiology [46] to study issues such as thermal stress, heat tolerance or thermal performance [55,59,60] in intertidal environments [40,41,55]. For example, ref. [61] measured mussel body temperature over 9 years in order to explore the relationship between body, air, surface and water temperatures and the spatio-temporal pattern of physiological stress. The data collected also provide a better understanding of the relationship between species activity rhythm, thermo-hydro requirements and environmental conditions [49,62].

### 1.3. Integration of Spatio-Temporal Variability in Mountain Biodiversity Monitoring Still an Overlooked Issue

As surprising as it may sound, compared to other ecosystems, there are not many papers regarding the conservation of mountain biodiversity [63,64], but several specific initiatives target efficiently these issues. Among them, the Global Mountain Biodiversity Assessment (GMBA) is a platform for international and cross-disciplinary collaboration on the assessment, conservation, and sustainable use of mountain biodiversity (https://www.gmba.unibe.ch/), while The Global Observation Research Initiative in Alpine Environments is a contribution to the Global Terrestrial Observing System (GTOS) aiming at establishing a long-term observation network for the comparative study of climate change impacts on mountain biodiversity [65]. Mountains are home to outstanding biodiversity, a great diversity of species, many of which are endemic [58,66,67]. Climate change induces altitudinal movements leading to changes in biotic interactions at each altitudinal level [68]. Changes in human land use, such as abandonment and fragmentation also conduct to imbalances in communities assemblages, host–parasite and prey–predator systems [58,68]. Furthermore, in most cases, induced upwards movements and land use entail a reduction in habitat area and an increased risk of extinction [69,70]. In addition, the warming is accelerated at high-altitude due to changes in snow albedo, water regimes and radiations fluxes [71]. For all of these reasons, mountains are priority habitats for implementing conservation policies [58,69,71]. However, mountain biodiversity knowledge is scarce, mainly due to the constraining field conditions [70]. The sites are difficult to access, isolated and remote. Moreover, most importantly, biodiversity studies must be replicated in space and time in order to integrate the spatio-temporal variability effect [72]. This type of study takes time, evidently even more so for those conducted over the long term. Thus, thanks to its ease of deployment, its remote recording capacity and the possibility of the sampling plan reproducibility over different

years, biomimetic logger seems to be an appropriate and powerful tool for the mountain biodiversity spatio-temporal study. Nonetheless, this tool is little used in this context. We found only one study that used biomimetic loggers in mountain and matched the constraint of addressing statistical issues of variability [62]: it illustrates the variation in activity time for lizard species according to altitude but only considered spatial variability, discarding temporal variability although being equally important.

Of all mountain species, ectotherms including reptiles as the Lacertidae family, are among the most affected species by climate change, making them indicator species [73,74]. Due to their thermal physiology conservatism, lizards are particularly sensitive to temperature changes [75,76]. Some adaptive responses can be implemented, for example, the selection of individuals with higher thermal preferences [77]. Thus, the lizards study provides a better understanding of how climate influences the activity rhythm and the persistence of ectotherms [78].

### 1.4. Activity Time and Persistence of Lizards

Unlike endothermic species such as mammals, which regulate their body temperature through their metabolism, ectothermic species, such as reptiles, control it through their behavior and physiology [78,79]. The thermal physiology of an ectothermic species can be modeled using a thermal performance curve. It describes performance over the tolerance range bounded by the critical lethal temperatures, CTmin and CTmax [80]. Development, survival, and metabolism can only occur within the tolerance range, which leads to thermoregulatory behavior [16,80]. In order to maintain body temperature within this range to optimize physiological processes, lizards alternate between outdoor activities such as basking, hunting, breeding, and retreating to their shelters [78,81]. The recording of a body temperature proxy using biomimetic loggers allows to qualify the activity pattern of the studied species [11,82].

The tolerance range shapes the activity pattern of individuals [11,83,84]. Face to climate change, the most observed response is the daily activity pattern change [79]. To buffer temperature increases, activity periods in the morning and late afternoon are reduced [82]. Individuals spend more time in shelters to avoid overheating and the activity time to gain energy and meet a sexual partner is shorter [11,82]. This response is effective, mere to implement but could become inadequate in the long-term, leading to a decline in fitness [85,86]. As reproductive success is positively correlated with activity time and energy budget, a reduction in activity time leads to an increased risk of local extinction [13,87–89]. Activity time over the critical breeding period could be used as a proxy for assessing the persistence of lizard populations [64,90]. However, the link between activity time and population persistence is never clearly explained in the papers. For example, the authors of [11] have defined either persistence or extinction using an activity time threshold. This method used in many papers [63,88] only indicates an on/off process whereas extinction is a continuous process. Here, we define a novel persistence index to qualify the conservation status of populations.

### 1.5. Study Aim

As previously demonstrated, variability, whether spatial or temporal, is insufficiently taken into account in biodiversity studies, even though it plays a major role in the implementation of biological responses to climate change. Our objective is to quantify the spatio-temporal variability effect in mountain environments on the activity time and persistence index of an endemic lizard species and to demonstrate the importance of its consideration to lead more appropriate conservation policies. We have made the following assumptions. Activity times and persistence indices would differ depending on exposure and elevation of the study sites. These two metrics could also vary between and within sites due to the habitat and micro-climate generated by the micro-habitat. Finally, climate variability would lead to inter-year variability. This environmental diversity could provide refuge locations where the micro-habitat would buffer rising temperatures. To achieve these, we measured the operative temperature using biomimetic loggers deployed over several sites and years, then we computed activity time and persistence indices and conducted statistical tests.

Given the context, the novelties of our study are the spatio-temporal study in the mountains thanks to biomimetic logger deployment—an original remote sensing tool for recording data specific to biological organisms—with the biodiversity conservation objective and the definition of a new persistence index.

First, we present the study area and the biomimetic loggers deployment sites. Second, we introduce the methods to calculate the activity time and the persistence index. Third, we discuss spatial variability on various levels (intra-site, inter-slope, inter-altitude, and inter-site) and inter-year temporal variability. On the basis of our study case, we demonstrate the interest of multi-site and multi-year studies. Finally, we provide guidelines for the use and deployment of biomimetic loggers.

## 2. Materials and Methods

### 2.1. The Pyrenean Rock Lizard (Iberolacerta Bonnali) as a Study Case

The Pyrenean rock lizard, *Iberolacerta bonnali* [91], is an endemic strictly rocky species living in Pyrenees between 1550 and 3144 m, although observations below 2000 m are scarce [92]. We studied populations present at three sites in the French Pyrenees in the Aspe-Ossau massif, called Peyreget, Anglas, and Arrious. Annual monitoring carried out since 2017 by the Climate Sentinels program indicates that local population density is at its lowest at the Peyreget site, which is the western edge of its known distribution, compared to other sites which have medium (Anglas site) or high density (Arrious site) (see in [93]). This small lizard measures 6.5 cm from the nose to the cloaca. Its activity period begins as soon as the melting snow in mid-June for the lowest sites and ends as soon as the first snowfalls in October, it hibernates for the rest of the year [92,94]. The critical breeding period occurs at the beginning of the active period. Thus, breeding success is highly dependent on environmental conditions, such as the insulating snow depth during hibernation, the melting period, and the temperature range at the end of hibernation. This small lizard is classified as Near Threatened on the IUCN national red list [95] and Endangered on the New Aquitaine regional red list [96], making it a priority species for biodiversity conservation studies. The species is mainly threatened by population fragmentation due to human activities and climate change [93]. The climate will become warmer and drier [97] which may increase the hibernation metabolic costs and consequently reduce the population fitness since the individuals reproduce at the beginning of their active period. Climate projections for the Pyrénées-Atlantiques show a continuous annual warming, from +0.2 °C to +0.3 °C per decade, until 2050. In terms of snow cover, that acts as thermal insulation during hibernation, future projections indicate a slight decrease in snow depth (from 2 to 3 days of snowfall) at 1800 m on the Aspe-Ossau massif until the 2030s, then a clear acceleration from the 2050s. According to the most pessimistic scenario (RCP 8.5), global warming could reach 4 °C by 2071–2100 and snow could become scarce by the end of the century [97]. Moreover, the reproductive characteristics of Pyrenean rock lizard, late age of sexual maturity (4–5 years), only one clutch per year, and low number of eggs (3 to 4), make it particularly vulnerable [94]. Like all ectothermic species, its activity rhythm is defined by its physiological characteristics (see Sections 2.3 and 2.4) and meteorological conditions, in particular air temperature.

### 2.2. Biomimetic Logger Deployment on Study Sites

In order to quantify the spatio-temporal variability of activity time and the persistence index of Pyrenean rock lizard populations, the study was conducted during the critical breeding period. Biomimetic loggers measured the operative temperature (Te), a proxy of the lizard body temperature if it remained at the location where the model was deployed [98,99]. In reality, individuals have developed adaptive thermoregulatory behavior to avoid temperatures that are too low or two high and do not stay in the same place [11,82]. This methodology makes it possible to quantify the possible activity hours, i.e., when the individual is not in his refuge and the inactivity hours constrained by body temperatures outside the thermal tolerance range [11,82]. Thus, it is very important to deploy the biomimetic loggers on lizard sunbathing areas, for example on screes rather than on the lawn for the

Pyrenean rock lizard. We illustrate this with a picture of another species present in the area, at lower altitude: *Podarcis liolepis* (Figure 1).

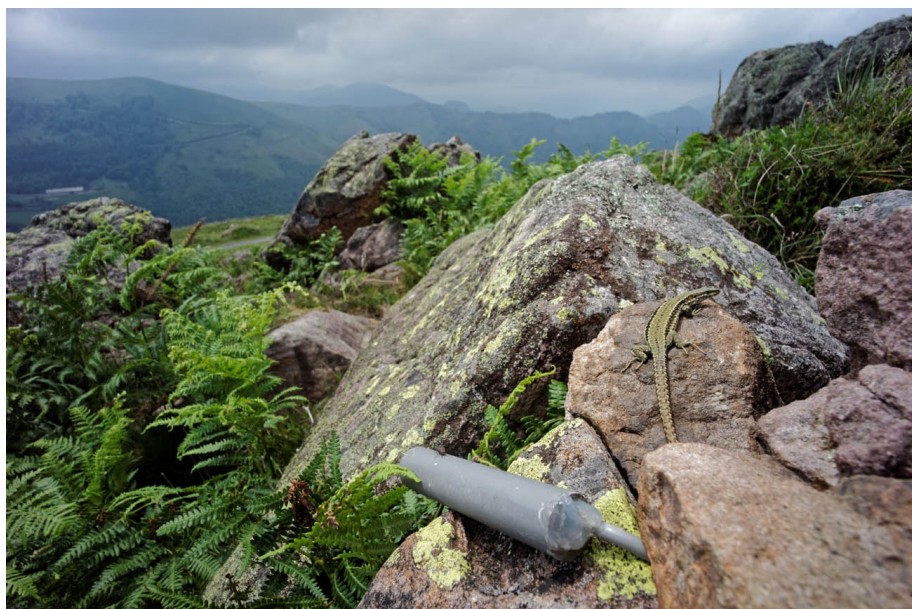

**Figure 1.** A PVC biomimetic logger at an exposure site, a *Podarcis liolepis* female basking nearby. The biomimetic model was placed on a sunbathing area. It is connected to the data-logger which is hidden under the rocks. Photographic credit: Matthieu Berroneau.

We used the HOBO U23-003 ONSET® data-loggers, which permit the measurement of outdoor temperatures as they are waterproof (https://www.onsetcomp.com/datasheet/U23-003). The sensors are thin which facilitates their insertion in biomimetic models and their operation range is −40 to 100 °C , which covers the Pyrenean rock lizard tolerance thermal range (CTmin = 6.1 °C and CTmax = 42.2 °C, unpublished work by Barry Sinervo, based on individuals captured at the Anglas site). Data-logger's thermal probe were inserted inside biomimetic models, PVC tubes the same size as a lizard and with the same thermal properties, the set is the biomimetic logger (Figure 1).

The deployments at the three sites allowed the study of different variability levels. The inter-slope and inter-altitude variability was studied through the biomimetic loggers deployment at Peyreget site in 2017, 2018, and 2019. Four localities were selected, two at about 2280 m (low altitude (L)) and two others at about 2430 m (high altitude (H)). For each altitude pair, one was located on the south slope (S) and the other on the north slope (N). To study the intra-site variability, two replicates spaced only one meter apart were deployed at about 2050 m on the Anglas site during the same three years (An$_1$ and An$_2$) and on Arrious in 2019 (Ar$_1$ and Ar$_2$). A fifth biomimetic logger added in 2019 on Peyreget (P) at the same altitude as those of the Anglas and Arrious sites allowed the study of inter-site variability. Finally, deployment over several years enabled to quantify the annual variability (Figure 2).

The biomimetic loggers recorded Te during the breeding period, which starts at the earliest in mid-June on Anglas and Arrious and ends at the end of September. To collect data, unlike traditional remote sensing tools, it is necessary to connect the data-logger to the computer using a handheld station. Then, the recording is restarted and the biomimetic logger is put back in place [23]. We defined a common study period to all sites and all years. Here, we considered the breeding period from 15 July at 00:00 a.m.—the beginning of the activity period on Peyreget, the highest of the three sites—to 2 September at 11:50 p.m.—the recording on the locality of NL in 2017 had stopped on the 3rd. Data recording is performed at a 10 min frequency, which represents a good compromise between the fine capture of environmental variations and the material storage capacity. Thus, Te time series obtained are composed of 7200 measurements.

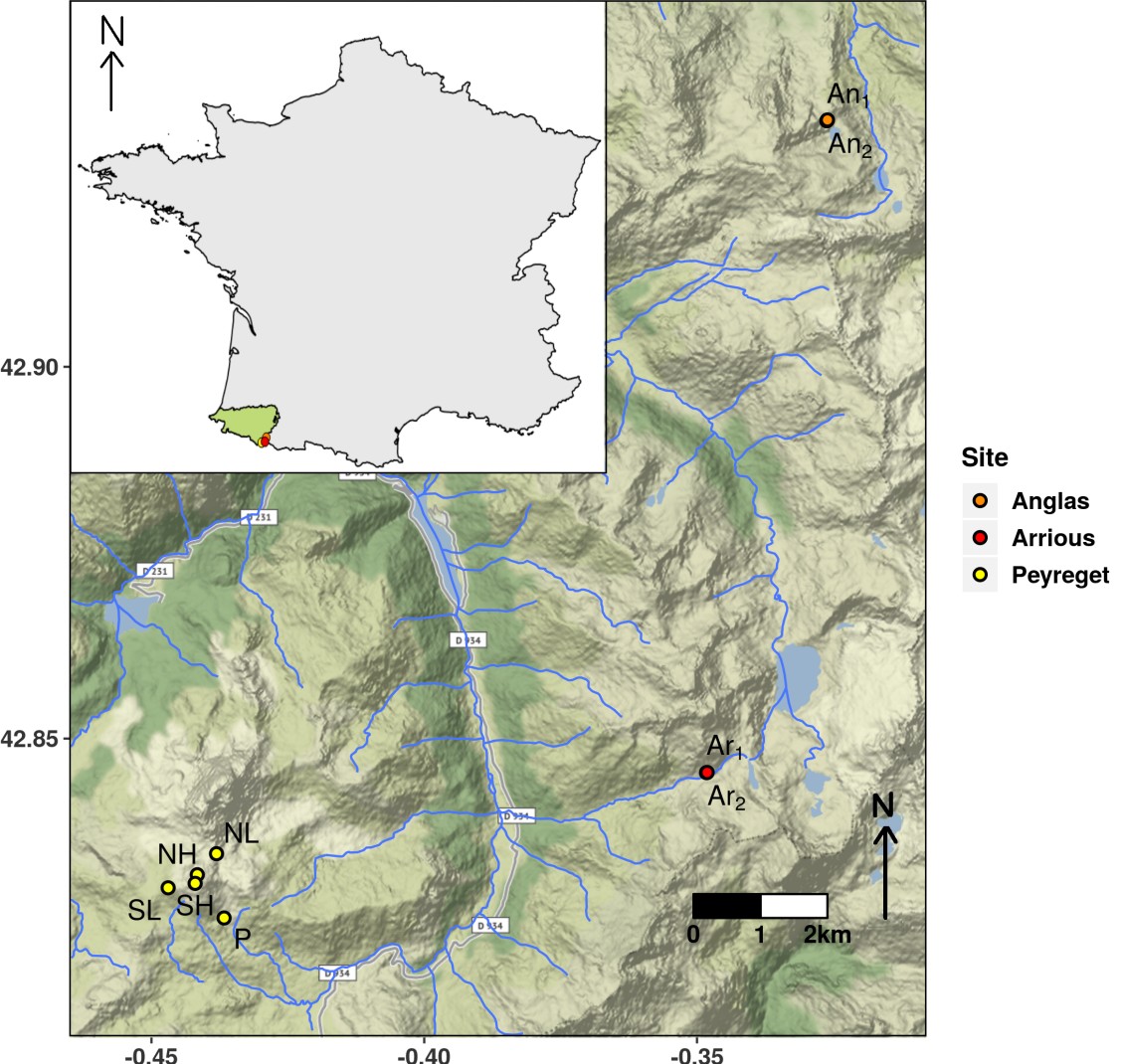

**Figure 2.** In France, in the department of Pyrénées-Atlantiques (in green on the France map in the top left), nine biomimetic loggers have been deployed on three study sites: Anglas—$An_1$ and $An_2$; Arrious—$Ar_1$ and $Ar_2$; and Peyreget—NL, NH, SL, SH, and P. The highest density is observed at the Arrious site (in red), the lowest at the Peyreget site (in yellow). The Anglas site (in orange) has a medium density. The hydrographic network is drawn in blue. Map source: Stamen, extracted with the get_map function on R.

### 2.3. Activity Time Computing from Te Measurements

Several methods have been developed to define the activity time and conversely the restriction time. Among the first ones, the restriction time can be defined when Te is greater than CTmax. However, this method suggests that individuals can be active even when body temperature is below CTmin [11]. Other authors explained that the restriction time is set when Te is above the upper limit of the preferred thermal range and the activity is limited by a lower value [63]. More recent studies have put forward a two-threshold method; the activity time is defined when Te is within the activity thermal range, bounded by the voluntary temperatures VTmin and VTmax, it is also called the voluntary thermal range (VTR) [83,84,100]. VTmin and VTmax are the 5th and the 95th percentiles of the preferred temperature distribution [83]. They also correspond to the lowest and the highest cloacal temperatures measured in the field on active individuals [84].

In this paper, we have calculated activity time using the two-threshold method because it appears to be more biologically relevant than the other and is more often used in recent papers [63,83] in line

with critical assessment by [84]. This method requires the definition of VTmin and VTmax for each of our study populations because the eco-physiological characteristics may differ between populations, as has been demonstrated for critical temperatures [101]. However, determining them experimentally would have been time-consuming and invasive. Other studies had been carried out on populations spatially close to ours, so we have favored the use of their results with the hypothesis that they presented similar thermal characteristics to the closest population studied in the bibliography [102]. We defined VTmin and VTmax using the [12] study conducted on the Blue Lake of the Bigorre Massif. This study was the best compromise between the geographical distance of our study sites and the availability of detailed data in the field. Thus, the VTmin was 20.8 °C and the VTmax was 35.2 °C for the three populations studied. For each measurement of Te occurring during the day, defined from sunrise to sunset, if included in the activity thermal range, the 10 min (the recording frequency) was defined as the activity time. Then, for each day, they were summed to obtain daily activity hours (HaDaily). Finally, for each recording (6 in 2017 and 2018; 9 in 2019), we summed daily activity hours over all days of the breeding period to obtain the total activity time (HaTot) (Equation (1)). For each breeding days *j*, daily activity hours is defined as

$$HaDaily_j = 10 \times HaCount_j / 60$$

where $HaCount_j = \sum_i \mathbb{1}_{\{Te_{ij} \in [VTmin, VTmax]\}}$, with $\mathbb{1}_{\{\cdot\}}$ being the indicator function and $Te_{ij}$ being the *i*th measurement of day *j*. Thus, total activity time is

$$HaTot = \sum_j HaDaily_j \tag{1}$$

*2.4. Persistence Index Computing from Overall Activity Time during the Breeding Period*

The definition of a persistence index allows to discuss the effects of spatio-temporal variability in terms of population viability and offers a more concrete interpretation of the variability than the activity times themselves. However, the link between activity time and population persistence is never clearly explained in the papers. The persistence or extinction definition in [11] study is based on the activity restriction times calculation at earlier periods at sites where extinction was observed today and where only climate change explained it. This method requires a very long-term monitoring of the sites and the confidence that only climate change is operating, so it could not be implemented in our study framework. Moreover, this method used in many other papers [63,88] does not provide a direct estimate of persistence probability or analogous proxy.

Here, we defined a persistence index (PI) based on a common persistence threshold to the three study sites (Tpersist). We assumed Tpersist represented the minimum total activity time required for recruitment at a site over a year for a certain breeding period. This threshold is calculated by multiplying the reference daily activity time (HaDaily$_{ref}$), i.e., the average number of daily activity hours, by the number of days in the breeding period. PI is the ratio of HaTot and Tpersist, it provides a point estimate, comparable within a site, between sites and between years. If it is within [0, 1], it represents the extinction probability (extP) and if it was greater than 1, it represents the activity time safety margin (HaSM), against the next increase in mean temperature and variability (Equation (2)).

$$\begin{cases} Tpersist & = HaDaily_{ref} \times \text{number of days in the breeding period} \\ PI & = HaTot / Tpersist \\ extP & = 1 - PI \text{ when } PI \in [0, 1] \\ HaSM & = (PI - 1) \times Tpersist \text{ when } PI > 1 \end{cases} \tag{2}$$

The persistence threshold can be defined on the basis of naturalist observations. For the Pyrenean rock lizard, according to field observations conducted on populations close to our study sites,

the HaDaily$_{ref}$ was almost equal to 4.5 h [12,92,103]. Thus, we defined the persistence threshold, Tpersist, as 4.5 h multiplied by 50, the number of days in the defined breeding period, i.e., 225 h for *I. bonnali*.

### 2.5. Statistical Analysis

As a prerequisite, the calibration of all sensors was performed by launching all the data-loggers at the same time and recording parameters of interest (temperature in our case) during a given period of time (usually three days) under the same controlled environmental conditions. The differences observed resulted only from the level of accuracy of the data-logger sensors, 0.21 °C for temperatures between 0 °C and 50 °C and the potential drift below than 0.1 °C per year (https://www.onsetcomp.com/datasheet/U23-003).

The HaDaily time series are compared in pairs to study the different variability types. Spatial variability is studied on intra-site (four tests), inter-slope (six tests), inter-altitude (six tests), and inter-site (eight tests) levels. Temporal variability is studied only at the inter-year level (18 tests). As the HaDaily distributions were not Gaussian and the 50 values of the time series were obtained for the same days for the two compared series, we decided to perform Wilcoxon Signed Rank bilateral tests for paired samples. To account for the zeros in the difference series, we used the *wsrTest* function from the *asht* package [104]. The estimated *p*-values are associated with the Hodges-Lehmann estimator and its 95% confidence interval. It corresponds to the pseudo-median of the serie of differences and is consistent with the median under the assumption that the serie of differences distribution is symmetric [105]. All *p*-values were corrected with the false discovery rate method, which is more powerful than Bonferroni to identify truly significant comparisons [106,107]. Finally, we plotted the transformed *p*-values to logarithmic decimal base and compared them to the 1.30 threshold corresponding to $-\log 10(0.05)$.

Persistence indices were compared in intra-site, inter-slope, inter-altitude, inter-site, and inter-year. We also calculated the PI inter-year mean and standard deviation to quantify the risk of error in the persistence estimates if the studies are conducted over only one year.

## 3. Results

### 3.1. Intra-Site Variability

All intra-site HaDaily comparisons on Anglas and Arrious showed significant differences (Figure 3). At the Anglas site (Figure 2), the pseudo-medians of the differences ranged from 0.75 h in 2019 to 2.08 h in 2017. These differences corresponded to one-sixth and up to almost half of the HaDaily$_{ref}$ of 4.5 h. These were reflected in the persistence indices, which differed from only 0.171 in 2019 (1.073–0.902) and 0.473 in 2017 (1.406–0.933). In 2018, the indices indicated the population persistence according to both records while in 2017 and 2019, one of the two indicated a non-zero extinction probability, respectively of 0.067 and 0.098. Regardless of the year, one of the replicates always had a higher persistence index than the other. At the Arrious site (Figure 2), the pseudo-median difference was 0.67 h and the extinction probabilities were 0.270 and 0.123 (Tables 1 and 2).

**Table 1.** Wilcoxon Signed Rank bilateral tests for paired samples on the intra-site level. *p*-values have been corrected by the FDR method and are shown in Figure 3. The pseudo-medians of the daily activity hours differences and their 95% confidence interval as well as the associated persistence indices (PI) are also presented. Significant differences are in bold.

| | *p*-Value | Pseudo-Median [IC$_{95\%}$] | PI |
|---|---|---|---|
| An$_1$ vs. An$_2$ in 2017 | $\mathbf{8.28 \times 10^{-6}}$ | $\mathbf{-2.08 \ [-3.00; -1.42]}$ | **An$_1$: 0.933, An$_2$: 1.406** |
| An$_1$ vs. An$_2$ in 2018 | $\mathbf{4.87 \times 10^{-4}}$ | $\mathbf{-1.67 \ [-2.42; -0.83]}$ | **An$_1$: 1.136, An$_2$: 1.510** |
| An$_1$ vs. An$_2$ in 2019 | $\mathbf{1.04 \times 10^{-5}}$ | $\mathbf{-0.75 \ [-1.00; -0.33]}$ | **An$_1$: 0.902, An$_2$: 1.073** |
| Ar$_1$ vs. Ar$_2$ in 2019 | $\mathbf{2.65 \times 10^{-5}}$ | $\mathbf{-0.67 \ [-1.00; -0.42]}$ | **Ar$_1$: 0.730, Ar$_2$: 0.897** |

**Table 2.** Interpretation of the persistence index, calculation of the extinction probability (extP), and of the activity time safety margin (HaSM).

|  | 2017 | 2018 | 2019 |
|---|---|---|---|
| NH | PI $\geq$ 1 → HaSM = 29.25 h | PI $\geq$ 1 → HaSM = 11.25 h | PI $\geq$ 1 → HaSM = 68.40 h |
| SH | PI < 1 → extP = 0.125 | PI $\geq$ 1 → HaSM = 34.65 h | PI $\geq$ 1 → HaSM = 7.20 h |
| NB | PI < 1 → extP = 0.250 | PI $\geq$ 1 → HaSM = 10.58 h | PI $\geq$ 1 → HaSM = 4.72 h |
| SB | PI $\geq$ 1 → HaSM = 33.30 h | PI < 1 → extP = 0.001 | PI < 1 → extP = 0.032 |
| P |  |  | PI $\geq$ 1 → HaSM = 43.88 h |
| $An_1$ | PI < 1 → extP = 0.067 | PI $\geq$ 1 → HaSM = 30.60 h | PI < 1 → extP = 0.098 |
| $An_2$ | PI $\geq$ 1 → HaSM = 91.35 h | PI $\geq$ 1 → HaSM = 114.75 h | PI $\geq$ 1 → HaSM = 16.42 h |
| $Ar_1$ |  |  | PI < 1 → extP = 0.270 |
| $Ar_2$ |  |  | PI < 1 → extP = 0.103 |

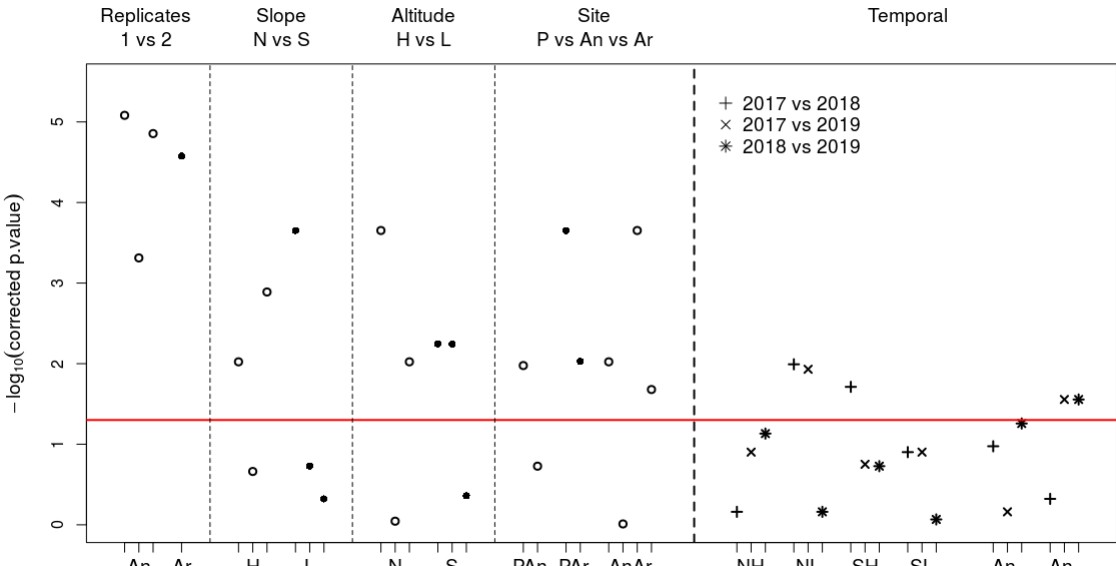

**Figure 3.** *p*-values of the 42 Wilcoxon Sign Ranked tests for paired samples, corrected with the false discovery rate method (FDR). The spatial variability (circles) is studied on the left side of the graph and the temporal variability (cross) on the right side. Concerning the intra-site variability (part Replicates) and the inter-slope and inter-altitude variabilities (parts Slope and Altitude), the three grouped graduations correspond to the analysis of the data from 2017 then 2018 then 2019. For the inter-site comparisons (part Site), the graduations permit to distinguish the two replicates from the Anglas and Arrious sites. The comparison order is the same as in Tables 1 and 3–5.

### 3.2. Inter-Slope and Inter-Altitude Variability

At the Peyreget site (Figure 2), the effect of slope was significant for half of the tests (Figure 3). At high altitude, the HaDaily pseudo-median was higher in the north than in the south in 2017 and 2019. At low altitude, the opposite was observed in 2017. The pseudo-medians of the differences ranged from 1.25 to 1.83 h, which corresponded to about one-third of the HaDaily$_{ref}$ of 4.5 h (Table 3).

The effect of altitude was significant on two-thirds of the tests (Figure 3). On the north slope, the HaDaily pseudo-median was higher at high altitude than at low altitude in 2017 and 2019; this result was also observed on the south slope in 2018. In 2017, on the south slope, the reverse trend was observed. The pseudo-medians of the differences ranged from 0.75 to 1.67 h, which corresponded to 17–37% of the HaDaily$_{ref}$ of 4.5 h (Table 3).

The persistence indices reflected these many disparities, ranging from 0.750 on NL in 2017 to 1.304 on NH in 2019. For five of the seven presented comparisons, only one of the two persistence indices was greater than 1 (NH vs. SH in 2017, NL vs. SL in 2017, NH vs. NL in 2017, SH vs. SL in 2017 and 2018), which also indicated a significant difference in the persistence probability. Finally,

four indices indicated an activity time safety margin greater than 10% of the persistence threshold of 225 h for NH in 2017 (+0.130 ~29.25 h) and 2019 (+0.304 ~68.40 h), SH in 2018 (+0.154 ~34.65 h), and SL in 2017 (+0.148 ~33.30 h) (Tables 2 and 3).

**Table 3.** Wilcoxon Signed Rank bilateral tests for paired samples on the inter-slope and inter-altitude levels. *p*-values have been corrected by the FDR method and are shown in Figure 3. The pseudo-medians of the daily activity hours differences and their 95% confidence interval as well as the associated persistence indices (PI) are also presented. Significant differences are in bold.

| | *p*-Value | Pseudo-Median [$IC_{95\%}$] | PI |
|---|---|---|---|
| NH vs. SH in 2017 | $9.43 \times 10^{-3}$ | **1.25 [0.50; 2.00]** | **NH: 1.130, SH: 0.875** |
| NH vs. SH in 2018 | 0.218 | −0.58 [−1.33; 0.25] | NH: 1.050, SH: 1.154 |
| NH vs. SH in 2019 | $1.29 \times 10^{-3}$ | **1.25 [0.58; 1.92]** | **NH: 1.304, SH: 1.032** |
| NL vs. SL in 2017 | $2.23 \times 10^{-4}$ | **−1.83 [−2.58; −1.00]** | **NL: 0.750, SL: 1.148** |
| NL vs. SL in 2018 | 0.187 | 0.42 [−0.17; 0.83] | NL: 1.047, SL: 0.999 |
| NL vs. SL in 2019 | 0.476 | 0.25 [−0.33; 0.92] | NL: 1.021, SL: 0.968 |
| NH vs. NL in 2017 | $2.23 \times 10^{-4}$ | **1.67 [0.83; 2.58]** | **NH: 1.130, NL: 0.750** |
| NH vs. NL in 2018 | 0.902 | 0.08 [−0.75; 0.92] | NH: 1.050, NL: 1.047 |
| NH vs. NL in 2019 | $9.49 \times 10^{-3}$ | **1.33 [0.42; 2.25]** | **NH: 1.304, NL: 1.021** |
| SH vs. SL in 2017 | $5.71 \times 10^{-3}$ | **−1.25 [−2.00; −0.50]** | **SH: 0.875, SL: 1.148** |
| SH vs. SL in 2018 | $5.71 \times 10^{-3}$ | **0.75 [0.33; 1.17]** | **SH: 1.154, SL: 0.999** |
| SH vs. SL in 2019 | 0.436 | 0.25 [−0.25; 0.83] | SH: 1.032, SL: 0.968 |

### 3.3. Inter-Site Variability

Inter-site comparisons between Anglas, Arrious, and Peyreget sites were carried out using temperatures recorded in 2019, and three quarters of the comparisons showed significant differences (Figure 3). The HaDaily pseudo-median on Peyreget site was higher than those calculated from the temperatures measured on Anglas and Arrious sites in three out of four cases. For comparisons between Anglas and Arrious, the HaDaily pseudo-medians on Anglas were almost always higher than those on Arrious. The pseudo-medians of significant differences ranged from 0.83 to 2.17, represented up to almost half of the HaDaily$_{ref}$ of 4.5 h. All of these differences were associated with persistence indices ranging from 0.730 for a replicate of Arrious to 1.195 for the P locality. For all comparisons except An$_1$- Ar$_1$, only one of the two indices was greater than 1 indicating differences in persistence between study sites (Tables 2 and 4).

**Table 4.** Wilcoxon Signed Rank bilateral tests for paired samples on the inter-site level. *p*-values have been corrected by the FDR method and are shown in Figure 3. The pseudo-medians of the daily activity hours differences and their 95% confidence interval as well as the associated persistence indices (PI) are also presented. Significant differences are in bold.

| | *p*-Value | Pseudo-Median [$IC_{95\%}$] | PI |
|---|---|---|---|
| P vs. An$_1$ | $1.06 \times 10^{-2}$ | **1.42 [0.42; 2.25]** | **P: 1.195, An$_1$: 0.902** |
| P vs. An$_2$ | 0.187 | −0.58 [−0.25; 1.42] | **P: 1.195, An$_2$: 1.073** |
| P vs. Ar$_1$ | $2.23 \times 10^{-4}$ | **2.17 [0.92; 3.17]** | **P: 1.195, Ar$_1$: 0.730** |
| P vs. Ar$_2$ | $9.37 \times 10^{-3}$ | **1.25 [0.42; 2.17]** | **P: 1.195, Ar$_2$: 0.897** |
| An$_1$ vs. Ar$_1$ | $9.49 \times 10^{-3}$ | **0.83 [0.33; 1.33]** | **An$_1$: 0.902, Ar$_1$: 0.730** |
| An$_1$ vs. Ar$_2$ | 0.975 | 0.00 [−0.50; 0.58] | An$_1$: 0.902, Ar$_2$: 0.897 |
| An$_2$ vs. Ar$_1$ | $2.23 \times 10^{-4}$ | **1.67 [0.92; 2.33]** | **An$_2$: 1.073, Ar$_1$: 0.730** |
| An$_2$ vs. Ar$_2$ | $2.09 \times 10^{-2}$ | **0.92 [0.17; 1.42]** | **An$_2$: 1.073, Ar$_2$: 0.897** |

### 3.4. Inter-Year Variability

Inter-year comparisons, made at Arrious and Peyreget sites, showed significant differences for only five of the eighteen tests (Figure 3). At NL, the HaDaily pseudo-median was lower in 2017 than in 2018 and 2019. At SH, the HaDaily pseudo-median was also lower in 2017 than in 2018. For one

of the two Anglas replicates, the HaDaily pseudo-median was lower in 2019 than in 2017 and 2018. The significant HaDaily differences ranged from 1.08 to 2.25 h, which corresponds to half of the HaDaily$_{ref}$ of 4.5 h. They were associated with persistence indices ranging from 0.750 to 1.510. At the NL and SH localities, the HaDaily difference was associated with a significant difference in persistence since the indices for 2017 indicated non-zero extinction probabilities, 0.250 for NL and 0.125 for SH. At Anglas, all indices involved in significant differences were above 1 and the activity time safety margin ranged from 16.42 h in 2019 to 114.75 h in 2018. Finally, the inter-year variability measured using the standard deviation ranged from $\pm0.096$ to $\pm0.228$ corresponding to $\pm21.6$ h ($0.096 \times 225$) and $\pm51.3$ h ($0.228 \times 225$) of activity (Tables 2 and 5).

**Table 5.** Wilcoxon Signed Rank bilateral tests for paired samples on the inter-year level. *p*-values have been corrected by the FDR method and are shown in Figure 3. The pseudo-medians of the daily activity hours differences and their 95% confidence interval as well as the associated persistence indices (PI), annual mean, and standard deviation are also presented. Significant differences detected are in bold, an asterisk indicates those that are also seem biologically relevant.

| | *p*-Value | Pseudo-Median [IC$_{95\%}$] | PI in 2017, 2018, 2019, Mean $\pm$ SD |
|---|---|---|---|
| 2017 vs. 2018 for NH | 0.690 | $-0.25$ [$-0.58$; 1.25] | 1.130, 1.050, 1.304, 1.161 $\pm$ 0.130 |
| 2017 vs. 2019 for NH | 0.125 | $-0.92$ [$-2.08$; 0.17] | |
| 2018 vs. 2019 for NH | $7.37 \times 10^{-2}$ * | $-1.08$ [$-2.33$; $-0.08$] * | |
| 2017 vs. 2018 for NL | **$1.02 \times 10^{-2}$** | **$-1.33$ [$-2.25$; $-0.42$]** | 0.750, 1.047, 1.021, 0.939 $\pm$ 0.164 |
| 2017 vs. 2019 for NL | **$1.17 \times 10^{-2}$** | **$-1.08$ [$-1.83$; $-0.33$]** | |
| 2018 vs. 2019 for NL | 0.690 | 0.17 [$-0.50$; 0.83] | |
| 2017 vs. 2018 for SH | **$1.94 \times 10^{-2}$** | **$-1.17$ [$-2.08$; $-0.42$]** | 0.875, 1.154, 1.032, 1.020 $\pm$ 0.140 |
| 2017 vs. 2019 for SH | 0.177 | $-0.75$ [$-1.58$; 0.17] | |
| 2018 vs. 2019 for SH | 0.187 | 0.50 [$-0.25$; 1.25] | |
| 2017 vs. 2018 for SL | 0.125 | 0.83 [$-0.17$; 1.67] | 1.148, 0.999, 0.968, 1.038 $\pm$ 0.096 |
| 2017 vs. 2019 for SL | 0.125 * | 1.08 [$-0.17$; 2.17] * | |
| 2018 vs. 2019 for SL | 0.857 | 0.08 [$-0.83$; 1.08] | |
| 2017 vs. 2018 for An$_1$ | 0.106 | $-0.75$ [$-1.83$; 0.08] | 0.933, 1.136, 0.902, 0.990 $\pm$ 0.127 |
| 2017 vs. 2019 for An$_1$ | 0.690 | 0.25 [$-0.67$; 1.08] | |
| 2018 vs. 2019 for An$_1$ | $5.53 \times 10^{-2}$ * | 1.25 [0.17; 2.17] * | |
| 2017 vs. 2018 for An$_2$ | 0.476 | $-0.50$ [$-1.83$; 0.83] | 1.406, 1.510, 1.073, 1.330 $\pm$ 0.228 |
| 2017 vs. 2019 for An$_2$ | **$2.78 \times 10^{-2}$** | **1.83 [0.42; 2.92]** | |
| 2018 vs. 2019 for An$_2$ | **$2.78 \times 10^{-2}$** | **2.25 [0.50; 3.67]** | |

## 4. Discussion

### 4.1. Why Should Multi-Site Studies Be Favored?

Previously, we use the term "site" to refer to a given location, namely, in our study, Anglas, Arrious, and Peyreget, three locations in the studied species habitat. In the following general section, the term "multi-site" refers to different spatial levels: that of replicates (intra-site comparisons), that of sites (inter-slope and inter-altitude comparisons), and that of the species' habitat (inter-site comparisons). Thus, in the following paragraphs, we detail the importance of considering these various levels.

#### 4.1.1. To Assess the Spatial Variability Due to Micro-Habitat

Contrary to what was expected, all comparisons between the two replicates placed only one meter apart showed significant HaDaily differences representing up to almost twice the reference daily activity time, HaDaily$_{ref}$. In addition, persistence indices widely varied, sometimes indicating a non-zero extinction probability and sometimes a persistence margin (at Anglas, PI varied from 0.902 to 1.510, Table 1). These results probably highlight the existence of the micro-climate felt by lizards at soil and scree level. The rocks composition and their density modulate the thermal conductivity and thus the micro-habitat temperature [108]. For instance, the Pyrenean rock lizard lives on screes composed of sandstone, shale, and volcanic rocks [94,109]. These rocks have different properties,

such as their mineralogy, granulometry, and water content, which influence thermal conductivity [110]. For example, thermal conductivity increases with the density of dry sandstone and siltstone rocks [110]. For granites composed of quartz and feldspath, minerals also present in sandstones, the authors of [111] demonstrated that thermal conductivity is higher when they contain water and have low porosity. Thus, all rocks in the Pyrenean rock lizard habitat have different thermal conductivities which may explain the differences observed in the calculations of activity times and persistence indices. The availability of various thermal micro-habitats can provide thermal refuges, which are essential to allow activity and contribute to population persistence [59,112]. Regardless of the year, one of the replicates always had a higher persistence index than the other which would indicate some consistency in micro-habitat quality. The spatial variability between two closely spaced data-loggers is rarely studied in the literature. The study of [113] dealing with the soil temperature variability showed that two data-loggers only 40 cm apart measured temperatures that differed by 4 °C on average. This study and our observations raise the importance of deploying a biomimetic loggers network in order to apprehend all the existing variability within the studied site.

### 4.1.2. To Assess the Spatial Variability Due to Habitat, Slope, Altitude and Hydrography

The inter-slope and inter-altitude study revealed a wide disparity in activity times and persistence indices. The pseudo-medians of the significant differences corresponded to between 17% and 41% of the HaDaily$_{ref}$ (at Peyreget, ranging from 0.75 to 1.83 (Table 3) in regard to HaDaily$_{ref}$ = 4.5 h). On 70% of significant differences, one persistence index was above 1 and the other below. Depending on the studied locality, the persistence index indicated the population persistence over all three years (NH locality) or only a few years (other localities). Thus, the study of a single locality is not representative of the study site [62] making spatial replication of sampling mandatory to measure the variability observed at the site and incorporate it into population management decisions [114].

For 70% of the detected differences, as expected, HaDaily pseudo-medians were greater in the north localities compared to the south and high compared to low. Over one on the three years, the activity time safety margins were also consistent with the expected, greater in the north and at high altitude than in the south and at low altitude. However, these patterns were not observed in the other years. For instance, in 2017, the daily activity times were higher at SL than at NL and SH and the persistence indices also reflected these differences. As the activity time depends on two thresholds, habitat suitability may be limited by too low or too high temperatures. In 2017, the percentage of operative temperatures (Te) included in the voluntary thermal range was only 24% for NL and 28% for SH localities while it was 37% for SL. Moreover, the percentage of Te above the VTmax was higher at NL than at SH than at SL, while it was more similar below the VTmin (Figure A1). These observations indicate that the activity time would be rather limited by too high temperatures. Slope orientation and habitat could explain these surprising results. Indeed, the in natura study did not allow to define sites strictly exposed to north and south and thus raised the important effect of slope orientation. Moreover, the scree diversity provides a thermal conductivity range, which is lower when the rock density is low and the stone blocks are distant (the scree type on SL) and higher when the scree is compact, wide, and broad (the scree type on NH and SH) [110]. Thus, a low elevation habitat may be more favorable than a high elevation habitat. All these results partially challenge the hypothesis that species would migrate to higher altitudes with more favorable temperatures [58,69] and highlight the role of slope orientation and habitat on the micro-climate felt by individuals [62,115,116].

Finally, the study of inter-site variability showed significant differences between sites at similar altitudes. In our study case, the HaDaily pseudo-median was always higher at the Peyreget site than at the Anglas and Arrious sites regardless of the replicate. Furthermore, HaDaily pseudo-medians at Anglas were higher than at Arrious. These differences are coupled with persistence indices indicating an activity time safety margin of almost 20% of the minimum total activity time required for recruitment at a site over a year (at the Peyreget site, PI = 0.195) while the indices indicate extinction probabilities for 3 of the 4 biomimetic loggers deployed on the two other sites (Anglas and Arrious).

However, these results seem to be at odds with the observed lizard densities as the Arrious site has the highest while the Peyreget site has the lowest. Here, the persistence index is only calculated from the activity time, which is itself only calculated from the Te, while other environmental parameters such as wind and humidity influence it and therefore the persistence [20,117,118]. At the Arrious site, local naturalists have already noticed that individuals were closer to the river when temperatures were higher (M. Berroneau and G. Pottier, personal communication) (Figure 2). This observation supports the new thermo-hydroregulation concept explained in [119]. A population close to a water source would be more tolerant towards high temperatures because it would benefit from higher air moisture that would reduce the dehydration risk [119]. Thus, the activity times and associated persistence indices may not reflect the observed lizard densities because the humidity was not taken into account. Moreover, the study of this particular site reveals patterns of more general importance and emphasizes the need of considering extra variables such as the distance to the nearest water point, air humidity or water balance to assess the population persistence and the species distribution [120,121]. It also highlights the relevance of studying several sites to characterize the variables combinations allowing the perenniality of a species [108,121].

### 4.2. Why Should Multi-Year Studies Be Favored?

Only 28% of the comparisons for the study of temporal variability showed a significant difference on HaDaily pseudo-medians while 71% were significant for the study of spatial variability (Figure 3). This would suggest that spatial variability would be more important than temporal variability. However, the largest undetected difference on tests of temporal variability is more than twice as large as on tests of spatial variability (1.25 versus 0.58, see Tables 1 and 3–5). Pseudo-median differences are therefore equally important for the temporal variability but they seem to be less easily detected. As underlined in [122], it would be appropriate to overlook the dichotomy between "significant" and "not significant" and focus on comparing results between them. Furthermore, an insignificant *p*-value may result from a large variability within the serie studied and does not necessarily mean that the tested hypothesis is true [123]. In our study case, on Anglas for the 2018 versus 2019 comparison, the pseudo-median is 1.25 h and is framed by the confidence interval [0.17; 2.17] (Table 5); this difference, although statistically insignificant after FDR correction, is biologically relevant. Furthermore, tests indicating estimators that correspond to almost 25% of HaDaily$_{ref}$ can be considered biologically relevant; here this concerns the differences 2018 versus 2019 on NH and 2017 versus 2019 on SL. Thus, there were appreciable biological inter-annual variability on all studied localities.

The pseudo-medians of the calculated difference series represented up to half of the reference daily activity time (from 1.08 to 2.25 h (Table 5) in regard to HaDaily$_{ref}$ = 4.5 h). These differences were associated with very various persistence indices depending on the year. For example, for the SH locality, the index indicated an extinction probability of 0.125 in 2017 and activity time safety margins of 34.65 and 7.20 h in 2018 and 2019 (Table 2). Conducting the study only in 2017 would have encouraged habitat protection policies while conducting it only in 2018 or 2019 would have indicated that populations would be maintained. Moreover, the average persistence index over the three years was 1.020 ± 0.140, which did not exclude the possibility of local population extinction in this locality. The same result was observed in the SL locality, the average index was 1.038 ± 0.096. Averaging persistence indices over several years allows a better assessment of the risk of local extinction. Indeed, if only one year is unfavorable, knowing that adults are capable of reproducing every year [92], this year will not be representative of the population fate. In our study case, in 2019, the heatwave from 22–25 July [124] was observed on all nine records, operative temperatures exceeding the upper critical lethal temperature CTmax at all locations studied except NH (Figure A2). Inversely, during the coldwave that occurred from 27–31 July [124], operative temperatures dropped below CTmin. Low altitude localities (~2050 m) were more impacted than the Peyreget peak localities (Figure A2). These excessively high or low temperatures force individuals to remain in their shelters for a longer period of time, which could

explain, for instance, lower activity times than in 2018 (Table 2). Thus, our results highlight the importance of considering inter-annual variability in biodiversity conservation studies.

*4.3. Remote Sensing for Biodiversity Conservation*

4.3.1. Biomimetic Loggers Development to be Even More Similar to Remote Sensing Tools

In the broadest sense, remote sensing tools allow to collect data about an object without contact with it [24,25]. Various methods can be used, such as reception of electromagnetic radiation emitted or reflected by the object [24] or use of a biomimetic model that mimics the physical properties of the object you wish to study coupled with a data-logger [11]. This second method permits the acquisition of animal-specific data which is not possible with classical methods. However, it requires at least the biomimetic logger deployment in the habitat of the studied species. The use of more expensive cloud data-loggers that send data to a server eliminates the need for data collection in the field (as done in our study) [125]. In this sense, coupling a biomimetic model with a cloud data-logger allows data to be directly obtained on a server as with conventional remote sensing tools, further facilitating multi-site and multi-year studies.

4.3.2. Linking Ecological and Remote Sensing Communities

Interdisciplinary approaches between biodiversity conservation and remote sensing fields are developing [30,126–128]. Remote sensing data, due to their great diversity, can be used for many applications [30]. For example, the plant specific diversity can be assessed from data characterizing the soil [129]. Maps produced by species distribution models are more reliable when the models integrate environmental, ecosystem structure or habitat quality variables [1,128,130]. Furthermore, remote sensing data allows to detect the effects of possible habitat modifications [54] and to follow the consequences of protection actions implemented [30]. Finally, the data collected in situ, for instance, physiological data from biomimetic loggers use or experiments, demographic data from counts, tracking movements data from transmitter collars, complement those acquired by conventional remote sensing methods and are essential for improving conservation strategies [127,131]. For example, defining the activity time (related to physiology) and the persistence index can allow the construction of a mechanistic species distribution model [88]. The activity time mechanistic variable can also be integrated in the same way as climate and habitat variables (possibly measured by remote sensing) in a distribution model [83]. This last modeling method, called the hybrid method, is growing rapidly because it combines the advantages of the two main distribution model types, the ease to use of correlative models and the explanatory power of mechanistic models [132,133].

**5. Conclusions**

Here, we highlighted the effect of the environment spatio-temporal variability on a body temperature proxy, the operative temperature, measured with an original remote sensing tool, a biomimetic logger. Relying on a level of replication as low as three sites over three years showed that proxies of activity times, calculated from operative temperatures and the associated persistence index, varied greatly at the all levels studied, i.e., within a single site, between sites, and between years. The newly defined persistence index provides interesting and promising basis to understand how different levels of variability may impact population sustainability but still needs formal validation through biological dedicated field or experimental studies. We encourage thus further research on the possible use of this method on other species and other temperature records, to gain confidence in its definition and possibly proves that it can become a new simple and cheap alternative to population studies. Getting back to remote sensing sensu lato, undoubtedly, biomimetic loggers, thanks to their large storage capacity, their ease of use, but above all their independent data collection on remote sites, are perfectly adapted to a multi-year and multi-site deployment. In addition, the deployment of this equipment enables noninvasive, reproducible, and low time-consuming studies. To sum up,

we advocate that biomimetic loggers find a place in the essential toolbox of anyone involved in biodiversity conservation programs, particularly for endangered species living in constrained habitats, such as mountain environments. To facilitate this appropriation, we provide below a guide for this tool use, addressing practical considerations based on our field experience. Finally, we highlight the lack of accessible statistical methods to study and compare the variability of interdependent data such as time series or the Earth's surface images, and encourage mathematicians to take up this underexplored research area.

## 6. A Quick User Guide for Biomimetic Loggers in Mountain Landscapes

### 6.1. Why Use a Biomimetic Logger/Sensor?

The analysis of spatio-temporal variability is essential and requires a lot of data to be collected in different locations and over several years. The greater the number of locations and years studied, the more accurately the different variability levels can be well studied. Massive data recordings require a reliable tool that is easy and quick to install, reasonably priced, and records data independently. Biomimetic logger remote technology offers new opportunities to improve the understanding of biodiversity responses to climate change by integrating the effect of spatio-temporal variability. It is a very suitable tool to better address conservation issues even in remote, hard-to-reach, and isolated habitats where monitoring is perilous [39,56]. Biomimetic loggers were broadly used for studies on the inter-tidal environment [40,55,59] and is developing in the mountain environment. Data recording only requires deployment and sometimes return to the field to collect them (see the last paragraph in Section 6.2 for more sophisticated alternatives), which greatly helps monitoring in these constraining habitats. Moreover, this tool ensures reproducibility of measurement methods [29] over each year and each study point and thus facilitates multi-year and multi-site studies. Thus, biomimetic logger could become an indispensable tool in the coming years hence the importance of knowing how to use them.

### 6.2. Materials

In essence, any biomimetic logger is composed of a data-logger whose probe is inserted into a biomimetic model [134]. Of course, the material should be chosen according to the parameter(s) one wishes to measure, e.g., humidity and temperature, and the characteristics of the probe, e.g., range, accuracy, resolution, and drift [65]. Whatever the kind of data-logger, the main challenge is to built a biomimetic model with same physical properties than organism [11,46]. To measure the operative temperature, a body temperature proxy, the biomimetic model has the same thermal characteristics as the animal in terms of radiation, conduction and convection and the data-logger is coupled to a thermal probe [46,55,61]. Similarly, to study the dehydration risk, the biomimetic model presents the water characteristics of the animal and a probe measures the relative humidity inside the model [135]. An infinite number of models can be constructed to answer a wide variety of biological questions.

To test the representativeness of biomimetic model, one usual approach is to compare data measured from the (physical) model and from the (biological) individual. For lizards, biomimetic models used are most often made of copper, PVC, or polypropylene [11,83,114]. To assess their reliability, a probe is inserted inside and the three types of models are placed in a vivarium with an individual equipped with a cloacal temperature sensor. At regular time intervals, temperatures are measured and then the four recorded series are compared [63]. This methodology permits to identify which type of model best represents the organism properties, with what accuracy, and to modify it if necessary.

In our study, like so many others (see, for example, in [11]), we use HOBO$^{\circledR}$ Pro v2 Onset U23-003 data-loggers with one or two temperature sensors. The environmentally friendly and robust housing of the HOBO Pro v2 data-logger is designed for years of reliable use in outdoor applications. Models U23-001 and U23-002 are also equipped with user-replaceable RH sensors. The recorder uses an optical USB communication interface (via hand-held or compatible base station) for launching and reading

the recorder. The optical interface allows the recorder to be discharged without compromising the electronics. USB compatibility allows easy installation and fast downloads. HOBOware® software version 2.2.1 or higher is required for operation of the recorder. For Onset HOBO® Pro v2 hardware, visit www.onsetcomp.com for compatible software. This type of data-logger is the simplest and most commonly used type: it requires only connection to the computer using a hand-held station to launch it and further connection to download the data [23].

There are more sophisticated types of data-loggers, such as those requiring a radio communication to transfer the data to the computer. The computer only needs to be close to the data-logger to collect the data, thus keeping the data-logger in place for the duration of the study. The battery lasts longer, accuracy and resolution are better. However, the radio channel is not available worldwide, which may hamper its use, and the cost is twice as high as the first type [65]. Other studies use "cloud data-loggers" [125], where data are sent directly to a server [136,137]. This type of data-logger has the enormous advantage of obtaining data in real time and thus being able to control the quality of the data and the correctness of the recording, but the cost of the equipment is much higher. It seems that they are (at least yet) not currently or widely used with biomimetic models.

*6.3. Prerequisites for the Installation of the Equipment in the Mountains*

Before installing the equipment in the field, especially since it is remote and the slightest omission will be detrimental, it is advisable to prepare and program it carefully before starting the field work. You will need the installation program, a recording device on your computer, and access to the internet. For instructions specific to the different types of recorders, the guidelines can be downloaded from the methods section provided on the manufacturers' websites or by consulting the help files for each type of instrument. Alternatively, or additionally, you can consult the websites of the major programs that have been successfully using this type of equipment for several years, such as the GLORIA program (www.gloria.ac.at), which offers in its "Methods" section (https://gloria.ac.at/methods/dataloggers) with very practical information on how to set up temperature recorders.

Important points to be checked include the following:

Time settings: The internal time of all temperature data-loggers must be set to GMT/UTC. To do this, check your computer's time to GMT/UTC when starting the loggers.

Start Time: When starting the loggers, choose "Delayed Start" and set the start time of all data-loggers to the same time. If all the temperature data-loggers used in your program start recording at the same time, you will allow direct comparisons and make data processing much easier. In our study, the recordings all started at 06:00 p.m. with a 10 min frequency, which allowed to characterize the time series with the same argument "start=c(1,109)", since 06:00 p.m. corresponds to the 109th value recorded over the day ($6 \times 18 + 1$).

Recording interval: The recording interval must be consistent with that corresponding to the ecophysiological quantity of interest. Allow for a safety zone of a few days to a few weeks and make sure that the recorder configuration is set to stop recording after this time or when its memory is full.

If the installation is to be carried out in places with special (private) property or conservation status (e.g., National Park) where special regulations apply, it is essential to obtain the necessary permits well in advance of the study.

As for the practice of mountain leisure activities as a space of freedom which requires constant vigilance and is subject to objective risks, the usual precautionary instructions concerning the practice of mountain activities must be observed. The weather conditions of the moment must be perfectly known and anticipated. The equipment must be adapted to the route envisaged on a varied terrain. Preparing, revising if necessary, the use of the appropriate equipment for the difficulties of the route is

an additional guarantee of safety. It is also a matter of course to inform your professional entourage or, failing that, your family and friends of the course you are planning to set up. All of these precautions will allow you to make the exit in the best conditions and will minimize the risk of an emergency call in the mountains.

*6.4. Installation*

Once in the field, you need to know how to install the equipment. The procedure obviously depends on the points discussed in Section 6.1. In our case study, we measured operative temperatures of lizards in mountain scree. In this case, the simplest installation consisted of placing the PVC tube imitating the surface lizard on a rock (see Figure 1) and hiding the case of the data-logger in the scree, taking care to protect the connection cable, which is vulnerable in an environment as mobile and exposed as a high mountain scree. A variant allows to measure not only the operative temperature outside as before, but also the operative temperature of a lizard hidden in the scree, mimicking the conditions prevailing in the rocky outcrop (sheltered from the heat); in this case, it is advisable to use a HOBO® U23 Pro v2 Onset data-logger with two External Temperature sensors. For the study of amphibians, where water loss may be of equal or greater importance than the thermal aspects, the biomimetic models used may be physical models in agarose (other possibilities exist, as discussed in [138,139], for example) coupled with data-loggers recording both temperature and relative humidity such as HOBO® U23 Pro v2 External Temperature/Relative Humidity Data-Logger; they allow to estimate the frequency of exceeding the limits of the thermal optimum (Topt) in conjunction with the critical water loss by evaporation (EWLcrit) as recently done by [135].

Depending on the biological question and the number of available biomimetic loggers, the deployment can be carried out in one or more occasions, over more or fewer sampling points. All biomimetic loggers can be deployed in the field and data will be collected only once at the end of the study [11]. It is also possible to deploy them on a sampling point for a certain time period, then collect the data and move them to another point and so on [140]. Moreover, they can be deployed always on the same micro-habitat, exposed rock to the sun [11] or on different micro-habitats, exposed and shaded rocks, different shelters [5,63]. The method of [140], which is slightly more constraining to implement, could allow the study of a larger number of habitat types while having a low number of biomimetic loggers. However, if it is necessary to obtain the data over the same time period, the first will be preferred. A biomimetic logger network allow to apprehend all the existing variability felt by species within the studied habitat [113]. Massive deployment will also minimize the dependency between the biomimetic logger location and the measurements [83].

NOTA: Design considerations are not addressed here; they are linked to issues that can be turned into questions beginning with the words: How much? When? Where? These very important aspects fall within the concepts of protocol and experimental design, which are covered in the specialized manuals. This field of knowledge is evolving very quickly. It is important to remember that it is essential to use probability sampling designs, without which data processing may be difficult and/or give biased estimates that prohibit or render abusive any inference [141–144]. Where there is no constraint other than to ensure the most spatially balanced distribution across study sites, there are spatially balanced sampling designs that are particularly useful [145].

Regardless of the probabilistic aspects underlying the design of the installation protocol, common sense must be respected with certain situations to be avoided as far as possible: peaks and edges of trails that are too crowded with tourists and hikers, areas subject to periodic maintenance fires, mountain areas that concentrate livestock or with an increased risk of trampling, cliff edges, crests exposed to strong winds, etc. These general recommendations are well known and detailed in the manuals of the major mountain research programs (in particular the GLORIA program, https://gloria.ac.at/downloads/manual [65]).

On a practical level, once the deployment strategy is decided, it is necessary to note for each deployed model, which biomimetic model is associated with which data-logger and which probe,

the sampling point GPS coordinates, the deployment dates and a distinctive element allowing it to be easily found. Any useful micro-habitat characteristics are recorded, according to the objectives of the program (see, for example, the survey sheets adopted within the GLORIA program). Whenever justified or imposed by conditions, the equipment should be secured by connecting it with a wire (for example) to the substrate. Finally, the device installation should be as discreet as possible to limit the risk of damage or theft of the equipment. We advice to keep the same biomimetic loggers each year on each site to get rid of the variabilities related to the measurement equipment. If the environment in which the material is installed makes it very difficult to detect the material (e.g., scree, rhododendron, or other shrub) it may be prudent to mark the site naturally (too much discretion may be ineffective) by means of a cairn, or artificially (sometimes too visible) by means of a colored biodegradable string tape, avoiding plastic or metal. Before, during, and after the installation, it is important to take pictures of the sampling point and the deployed material. Field sheets must be properly constructed to allow the collection of all this information, we can rely on the many protocols developed by the GLORIA program [65]. Photographic documentation is crucial for various reasons, for instance, to visually document the whole and details of the installation, to find the material at the end of the recording period (which is not obvious in a mountain scree for example) and for the precise reallocation of plots if necessary.

### 6.5. Field Recording and Its Constraints

By following the advice developed above, and in particular by avoiding exposed or disturbed environments (hiking, periodic maintenance fires, cliff edges, hilltops exposed to strong winds, etc.), recording will take place in the best conditions. However, various other hazards, most of which cannot be controlled, are bound to occur. They are due to particular weather conditions (heavy rain or lightning) or the activity of organisms present on the site (for example, the foraging activities of small mammals or the curious behavior of birds). By respecting the recommendations listed in Sections 6.3 and 6.7, the risks of loss of recording quality must be reduced as much as possible.

To date, most biomimetic loggers use devices that has to be connected to a hand-held station or a computer to download the data, as they are cheaper. The risk with that kind of equipment is that battery suddenly collapses or the biomimetic model (i.e., the PVC tube) is moved by a person or a foraging animal: on both cases, there is no way to know that something went wrong. More sophisticated cloud data-loggers allow for permanent control of the quality of the data and the correctness of the recording but their cost being higher.

### 6.6. Data Processing

The quality of the recorded data can be checked by representing the data and calculating basic statistics such as quantiles, mean, and extremes. Then, the statistics should be discussed with the data from the literature to assess their consistency. Data can be corrected by various programs in which quality criteria are indicated [146,147]. The use of the data-logger type requiring field feedback does not allow real-time control of the data, but their operation is rather reliable. In our study, all the time series had consistent values and only one of them (over 21) was truncated: the cable connecting the probe to the data-logger was torn by being gnawed by a foraging mammal. To overcome this problem, the equipment was subsequently wrapped with wire mesh in 2018 and 2019. Thus, this type of data-logger represents a good compromise between cost and reliability.

### 6.7. Maintenance and Storage of Equipment

With proper maintenance, the equipment should provide years of accurate and reliable measurements. In the case of a study conducted over several years, there are two scenarios: either the monitoring is continuous or the monitoring is discontinuous. In the case of a continuous installation, the equipment is installed permanently for the duration of the study, and in this case there is no maintenance or storage as such. However, depending on the frequency of data

collection, two points should be checked. First, the functioning of the equipment in place—in general, the flashing of a led attests to this, but one must be vigilant when reading it, especially in full light, and be patient, the frequency of flashing is sometimes irregular and may differ according to the data-logger. Second, the level of residual energy—it is essential to check the batteries, or change them as a precaution, especially if the monitoring time is longer than the life span declared by the manufacturer (for the Onset HOBO® Pro v2 U23-00x series, see for example: https://www.onsetcomp.com/files/manual_pdfs/10694-H-MAN-U23.pdf). In the harsh mountain environment, especially with low winter temperatures, battery life is shortened.

In the case of a discontinuous installation, the equipment is installed several times (and therefore removed as many times) throughout the duration of the study, and in this case there is minimal maintenance and storage precautions during the period of non-use in the field. This is the case, for example, in a study conducted over several years, with deployment of equipment only during each breeding period. In the case of our study, conducted from 2017 to 2021 in the context of the program "Climate Sentinels" (https://www.sentinelles-climat.org/), equipment was deployed only during the breeding period of the species concerned; over the five years, equipment was thus installed and removed five consecutive times. It is more prudent, when the technical configuration of the equipment allows it, to disassemble the simple elements and dry them carefully (without, of course, touching the connectors and the electronic circuits). Some types of equipment, for example the HOBO® Pro v2 Onset with one or two temperature sensors that we use and more generally the HOBO® Pro v2 U23-00x Onset series, contain bags of desiccant (Silicagel® type), which can be changed before each new installation or dried (oven or hair dryer). Between each deployment period, the equipment is stored under the conditions prescribed by the manufacturer, which correspond at least, as common sense dictates, to average temperatures in a dry place. It is more prudent to remove the batteries during this period to avoid problems associated with maintaining batteries in any type of equipment. If the equipment has markings, it is useful to check that these are legible and if necessary to renew them. In the case of the HOBO® Pro v2 External Temperature/External Temperature Data Logger Onset hardware we use, the data-logger wire connecting the probe (placed inside the biomimetic model) to the case of data-logger will be examined and repaired (e.g., taped) if traces of injury or bites appear on these wires; an alternative is to change the wire when possible.

Last but not the least, quality control is key to the success of a program. Before each deployment period, the reliability of the data-logger probe is checked. Quantity recording by the data-logger and the associated probe can be launched in a controlled environment where the quantity is also measured by another reliable probe. The comparison of the two series obtained then allows to identify any possible errors in the recordings by the data-logger–probe couple tested as well as the drift [63].

**Author Contributions:** Conceptualization, F.D.; Methodology, F.H. and F.D.; Software, F.H. and B.L.; Validation, F.H., F.D. and B.L.; Formal analysis, F.H.; Investigation, F.H. and F.D.; Writing—original draft preparation, F.H. and F.D.; Writing—review and editing, F.H., F.D. and B.L.; Visualization, F.H. and B.L.; Supervision, F.D. and B.L. All authors have read and agreed to the published version of the manuscript.

**Funding:** This research was funded by the Climate Sentinels Program, carried out in New Aquitaine, France, from 2016 to 2021. The project is supported by the European Union within the ERDF framework, the New Aquitaine region and the Atlantic Pyrenees and Gironde departments.

**Acknowledgments:** We thank all funders and partners of the Climate Sentinels program (coordinator: Fanny Mallard). In particular, we would like to thank Matthieu Berroneau of the Cistude Nature association (director: Christophe Coïc) for sharing his naturalist knowledge. Also, we thank the Pyrenees National Park for the authorizations to deploy the biomimetic loggers obtained in 2017 (n° 2017-228), 2018 (n° 2018-99) and 2019 (n° 2019-121).

**Conflicts of Interest:** The authors declare no conflict of interest. The funders had no role in the design of the study; in the collection, analyses, or interpretation of data; in the writing of the manuscript; or in the decision to publish the results.

## Abbreviations

The following abbreviations are used in this manuscript.

| | |
|---|---|
| Te | Operative temperature |
| Ha | Activity time |
| HaDaily$_{ref}$ | Reference daily activity time |
| PI | Persistence Indice |
| CTmin and CTmax | Minimum and Maximum Critical lethal Temperature |
| VTmin and VTmax | Minimum and Maximum Voluntary Temperature |
| VTR | Voluntary Temperature Range |

## Appendix A

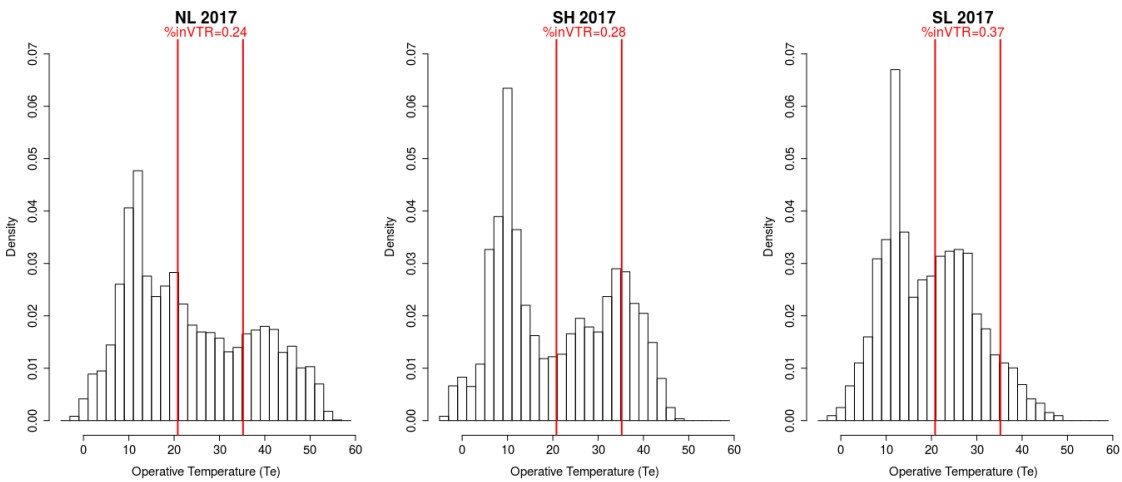

**Figure A1.** Operative temperatures (Te) measured over the day, defined by sunrise and sunset, over the breeding period. Percentage distribution of Te within the voluntary thermal range (VTR) also called the activity thermal range.

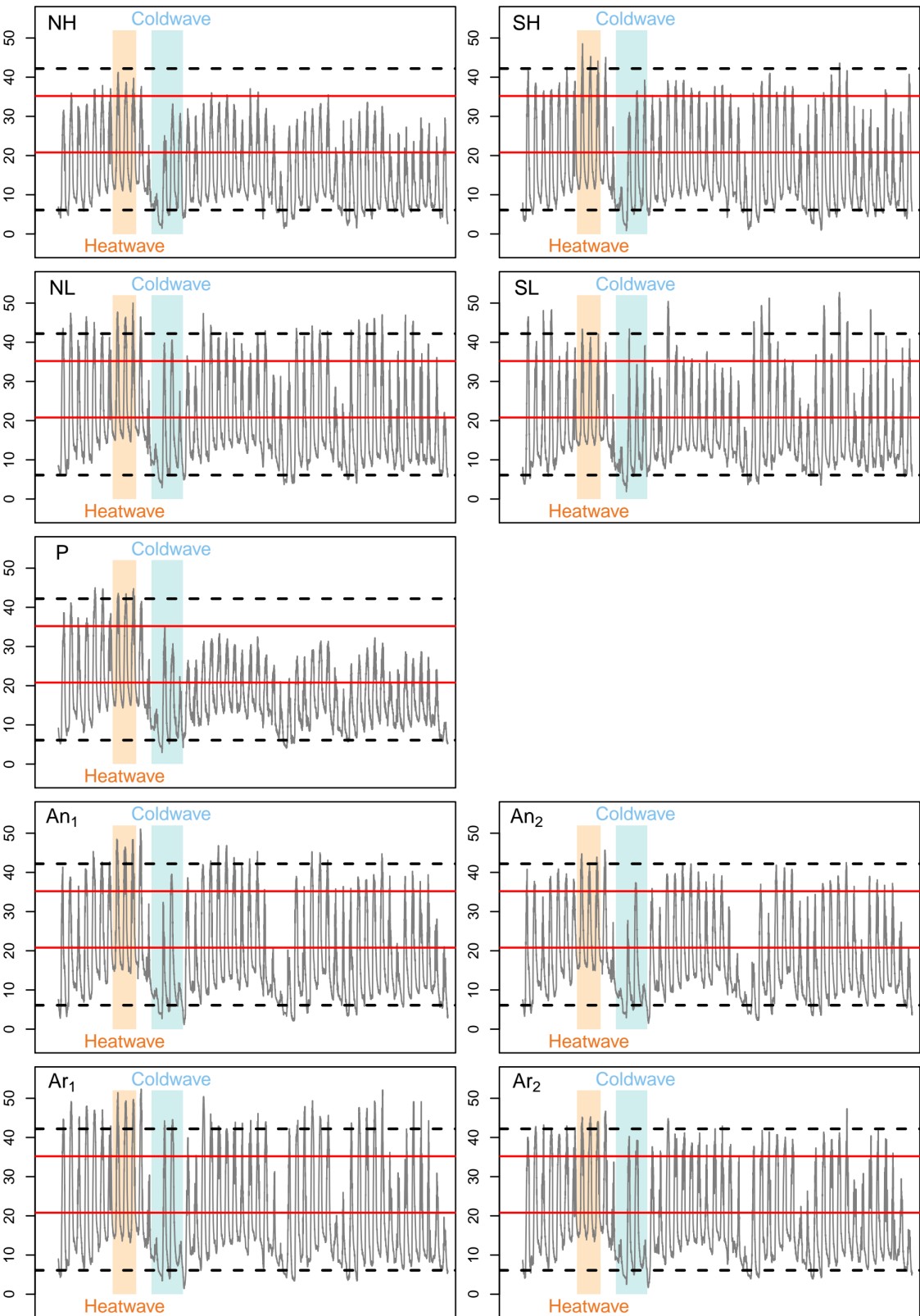

**Figure A2.** Operative temperatures time series measured on the different study localities in 2019 from 15 July to 2 September. The heatwave from 22–25 July (beige polygon) and the coldwave from 27–31 July (blue polygon) were observed on all sites. The red lines represent VTmin and VTmax which bounds the activity thermal range and the dotted black lines represent CTmin and CTmax which bounds the tolerance thermal range.

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
