# Peer review of "Multi-Site and Multi-Year Remote Records of Operative Temperatures with Biomimetic Loggers Reveal Spatio-Temporal Variability in Mountain Lizard Activity and Persistence Proxy Estimates"

_remotesensing, doi:10.3390/rs12182908_

Round 1

Reviewer 1 Report

Main comments

This manuscript used data-logger of temperature to study the activity of lizard in alpine biomes in Pyrenean Mountain. This study has very good results about the distribution and population development of endangered lizard. It has very little relation with remote sensing and great implication of this study to the ecological niche variability of lizard population (take home message) is still lacking.

General comments

Line 2 mean value is used more than other factors.

Line 12-13 The take-home message needs to be refined.

Line 24 extreme events could also happen frequently in spring.

Line 145 In this section, you should have a hypothesis based on the previous study.

Line 156-160 These climate data loggers have already been used in many aspects such as micro-climate and plant and animal ecology. You should first show the guideline of these data logger and then the main findings of lizard activity and so on.

Line 163-175: In method section, you need more information about the rock lizard. For example, do you have any reference about the exact activity of temperature threshold value about this species? Based on such information, your study is more meaningful using temperature data logger. Or you can calculate the temperature threshold based on your data.

Line 180-182 the population density is different between sites. Maybe you could add some discussion about the reason based on your results of activity time.

Line 177-189 Also, more information about the place where you install the temperature data logger. Whether the micro-climate of these sites represents the real habitat of rock lizard? The lizards are not always under the sunshine places. They are prone to stay at safe and shade sites in the night to avoid predators.

Line 191-193 Maybe you can delete this sentence because everyone know its measuring range according to the type.

Line 334-339 Everyone know that multi-site study is robust than single site. Maybe this paragraph is useless.

Line 375-376 This belong to the results. Please do not repeat the result in your discussion.

You should give some key information about the population development of lizards under the climate change.

Line 389-392 In your opinion, whether the lizards population move downwards in elevation?

Line 450-441 Maybe the regional climate information is needed in the method section. The readers want to know the background of climate (temperature and precipitation) trends. Warming and drying trend or warming and wetting trend?

Line 438-440 Are there any extreme temperature events in your study period? It is interesting to test the lizard population dynamics in response to extreme high or low temperature. I think your study area (Mediterranean region) is becoming warming and drying. Also, the extreme climate condition will become more frequent according to the IPCC report.

Reviewer 2 Report

The submitted manuscript entitled "Multi-site and Multi-year Remote Records of Operative Temperatures with Biomimetic Loggers Reveal Spatio-temporal Variability in Mountain Lizard Activity and Persistence Proxy Estimates" highlights the importance of biomimetic loggers (data loggers) for climate studies and their link to biodiversity research and how scientist shall be aware of the potential of such measurements in terms of spatial and temporal variability over complex natural environments. The presented article provides an extensive review, provides several references on the related subject, is well-written, and deserves publication.

My first impression by reading this manuscript for the first time (starting with the abstract only) is that the subject would better fit with the sister Journal Sensors. However, when reading the paper and reaching some sentences (for example from L54 to L56) I started to identify the possible link with remote sensing.

However, in general, this link is still poor and should be more evident in different parts of the paper. I enjoyed reading the paper and I found it worth for publication anyway, but this link would surely fit better to the goals of the Remote Sensing journal and also attract a larger audience.

I particularly found that it could be an interesting contribution to the remote sensing community given its novel contribution. In remote sensing, it is very important to link ground measurements that are also costly and time-demanding. Time-series studies are also very important and reveal aspects that are not possible with static and single remote sensing prints in a given year. In this sense, the nature of the remote sensing data and the physical mechanisms involved could be mentioned and better linked to each other.
Besides temperature, data loggers could be useful for other parameters that could also be mentioned. The link could be established explaining how the implementation of multi-year and multi-sites approaches using biomimetic loggers would be possible (if ever possible) with a given remote sensing systems/sensors (if orbital, airborne, terrestrial, or even proximal) and the challenges on doing it. Interestingly would be to mention which remote sensing systems/sensors would be feasible for that task and what would be the current or even forthcoming demands whether this is still not totally reachable yet.

Below some other minor concerns:
L4: please establish the importance of this study with remote sensing data;
L9-10: are those parameters modeled by remote sensing?
L153-155: a link must be better established with remote sensing;
L164: start to introducing your study area here;
Figure 1: please add a country and region map; it is not that clear what the color code means for population density, please clarify; whenever possible, please add the colors for the altitude range and also the DEM data source used in the caption; if there is also a drainage network available for this section please add it;
L221: what "biologically relevant" mean? please clarify;
L359: Check for guidelines, I guess it would be Picebourde and Salle [99] studied the soil…;
L403: which naturalists? in this area? please add references;
L460: please add a mention that such variability is a challenge not only for mathematicians but also for the remote sensing users in real complex areas;
L461: I would suggest adding this interesting material as an appendix;
L694: add the References section;

Round 2

Reviewer 1 Report

Authors have sufficiently addressed comments.